# Validating the diagnostic performance of MUAC in screening moderate acute malnutrition and developing an optimal cut-off for under five children of different regions in Ethiopia

**Abera Lambebo**[1]\*, **Yordanos Mezemir**[2], **Dessalegn Tamiru**[3], **Tefera Belachew**[3]

**1** Department of Public Health College of Health Science, Debre Berhan University, Debre Berhan, Ethiopia, **2** Department of Public Health, Debre Berhan Health Science College, Debre Berhan, Amhara Regional State, Ethiopia, **3** Faculty of Public Health, Department of Nutrition and Dietetics, Jimma University, Jimma, Ethiopia

\* lambebo70@gmail.com

**Data Availability Statement:** All relevant data are within the paper and its Supporting Information files.

## Abstract

### Background

Valid and reliable anthropometric indicator is useful for early detection and treatment for under nutrition. Although, mid upper arm circumference (MUAC) is used for screening of children with moderate acute malnutrition in Ethiopia, its performance for the different ethnic groups has not been evaluated.

### Objective

To determine the diagnostic performance of MUAC for determination of moderate wasting among children of different ethnic background and develop optimal cut-off.

### Methods

A community based cross-sectional study was conducted among under five children of the three regions namely: Somalia, Amhara and Gambella Regions. The diagnostic performance of MUAC was validated using weight for height Z-score< -2 as a gold standard binary classifier. Test variable is mid upper arm circumference (MUAC< 12.5cm) and weight for height Z-Score (WHZ) is standard variable. ROC analysis performed based on the assumptions of MUAC value lower the cut-off point indicates the undernutrition. Area under the curve and validity measures (sensitivity and specificity) was generated as parameter estimated. The results were presented using tables and ROC curves.

### Results

Except in the Gambella region, there was fair agreement between MUC<12.5cm and Weight for Height Z score<-2 in diagnosing wasting in Somali (Sensitivity = 29.3%, Kappa =

**Funding:** The authors received no specific funding for this work.

**Competing interests:** The authors have declared that no competing interests exist.

0.325, P<0.001) and in Gambella regions (Sensitivity = 16.7%, Kappa = 0.19, P<0.001). In Amhara region there was fair agreement between the two measures in diagnosing moderate acute malnutrition (MAM) (Sensitivity = 16%, Kappa = 0.216). For the Overall sample, the sensitivity of MUAC<12.5cm was 20.6% (Kappa = 0.245, P<0.001. Based on ROC analysis, the optimal cutoff value of MUAC for diagnosing moderate acute malnutrition for the two regions namely for Gambella and Amhara was 13.85cm with sensitivity of 0.99 and 1.00, respectively. However, for Somali Region the optimal cut was 13.75cm (Sensitivity = 0.98 cm and specificity = 0.71).

## Conclusion

Findings revealed that the inter reliability of measurement for MUAC< 12.5cm and WHZ<-2 for diagnosing MAM was low among different ethnic groups with the cut-off varying in each region. The existing cutoff point is less sensitive for diagnosis of MAM. As Ethiopia is home of diverse ethnic groups with different body frame and environmental conditions, the new cut off points developed for each region recommended to be used for screening moderate acute malnutrition to prevent relapse of MAM and reduce chronic malnutrition.

## Introduction

### Background

Malnutrition refers to shortages, excesses or disparities in a person's intake of energy and/or nutrients and it encompasses two broad groups of conditions over nutrition and under nutrition [1]. It is an imbalance between the nutrients the body needs and the nutrients it gets and over nutrition or consumption of too many calories or too much of any specific nutrient protein, fat, vitamin, mineral, or other dietary supplement, as well as undernutrition or deficiency [2]. One or more forms of malnutrition affect every country in the world [3]. Globally 50.8 million children (22.2%) under five years of age are stunted, 50.5 million children under five are wasted and 20 million newborn babies are estimated to be of low birth weight, while 38.3 million children under five years of age are overweight [4] and In Ethiopia 36.8% of children, under 5 years of age are stunted and 7.2% of children wasted [5].

   Nutritional status assessment can be done by using anthropometric, clinical and biochemical methods and, dietary practices [6]. Anthropometry is the measurement of the human body in terms of the dimensions of bone, muscle, and adipose tissue Anthropometric measurements are preferred methods widely used because they are economical and non-invasive measure of the general nutritional status of an individual or a population group [7]. In children, measurements reflect general health status, dietary adequacy and growth and development over time, while they are used to evaluate health and dietary status, disease risk, and body composition in adults [8]. Although height and weight-based anthropometric measurement is an excellent tool assess general nutritional status in a population [9], it is not used for practice at the community level due to issues related to cost, ease of measuring and transporting the instruments in geographically scattered and mountainous terrains. As a result, there is uncertainty among practitioners and academic specialists, concerning the consequences of screening children with severe acute malnutrition (SAM) solely using weight-for-height z score [10] in developing country's contexts.

Mid Upper arm Circumference(MUAC) is considered as a simple tool for the assessment of acute malnutrition in different ethnic back grounds [11]. MUAC is a decades-old anthropometric measurement of the amount of muscle in the arm, which theoretically reflects the total amount of muscle or protein in the body [12]. In programs managing acute malnutrition [13], MUAC bids the advantages of easy-to carry, even to geographically hard to reach areas because it needs slightest preparation and it is effective in the assessment of nutrition status when measured with care and precision [14,15]. This makes MUAC one of anthropometric tools as effective as the body mass index-for-age z score for assessing mortality risks associated with under-nutrition among African children and adolescents [11]. As MUAC is easy to utilize during Acute malnutrition in hard to reach are or area with limited resources and it is best fits in Africa as poor infra structure is common. According to the WHO 2009 children 6–60 months with a MUAC less than 125 mm are considered as under nourished [16].

Mid-upper arm circumference is easier to measure and interpret and it is similar in boys and girls and is relatively constant from 6 months to 5 years that and avoids the requirement to calculate exact age [17]. Study in Senegal suggests that MUAC is better than WHZ to identify high-risk children in the community and using both WHZ less than -2 and MUAC less than 125 mm increases specificity but decreases sensitivity to identify high-risk children. It was reported that there is no advantage of combining WHZ and MUAC to identify high-risk children [18].

However, there are doubts about sensitivity and specificity of MUAC<125 mm on identification of undernutrition among children implying an urgent need for revising the cutoff value of MUAC to improve its sensitivity and specificity based on locally and ethnically relevant data [19,20]. Another challenge related to MUAC is that it is based on a single cut-off value for all the children less than 5 years. However, it has recently been questioned whether MUAC is age-and sex-independent [10] or not and it is suggesting the gap to develop age and sex specific MUAC cut off point. In addition to that the currently existing cut-off point for anthropometric measurements have lower survival rate, increasing number of relapse and overburdening health system by relapse cases [21].

It may also be sensed inadequate nutritional status (growth) is not necessarily due to inadequate diet but also due to slightly genetic or ethnic and geographical variation that may make a difference in nutritional status and in the sensitivity and specificity of MUAC in detecting malnutrition. Therefore, this study aimed to determine the diagnostic performance of MUAC for determination of moderate wasting among children of different ethnic background and develop new optimal cut-off point.

## Methods

### Study area and design

Community based cross-sectional study was conducted among under five children of the three regions namely: Somali, Amhara and Gambella Regions of Ethiopia. Gambella region has three administrative Zones which is bordered by Sudan to the south, west and north, by Administrative Zone 1 to the east and Administrative Zone 2 to the southeast; Towns in this zone included Tergol and Telut. Most of the area of this zone was added to Nuer Zone and some parts were added to Anuak Zone and the third one Mejang Zone. For this study Gambella Town used to represent two ethnic groups those were Nuer and Anuak to represent Nailo Saharans' linguistic groups.

Amhara Regional state was another region that participated in this study. Amhara is bordered by the state of Sudan to the west and northwest, and in other directions by other regions of Ethiopia: Tigray to the north, Afar region to the east, Benishangul-Gumuz to the west and

southwest, and Oromia to the south. The region is subdivided into 10 administrative zones. Agew Awi, East Gojjam, North Gondar,North Shewa, North Wollo, Oromia,South Gondar, South Wollo, Wag Hemra, West Gojjam and Bahir Dar (special zone) for this study North Shewa zone was selected randomly to represent Sematic linguistic groups.

Somali Regional State is bordered by Afar and Oromia regions, and the chartered city Dire Dawa to the west, as well as Djibouti to the north, Somalia state to the north, east, and south; and Kenya to the southwest. Region is subdivided into eleven administrative zones and Six Special administrative zones: For this study Sitti Zone (formerly Shinile) was selected randomly from the other Zones to represent Cushitic group's linguistic groups.

The study population comprised of under five children from the three regions namely Gambella, Amhara and Somalia, Ethiopia. All apparently healthy under five children who were from selected regions were include in the study. Under five children who are from another ethnic background, those with deformity that interferes with anthropometric measurements were excluded and those who were severely ill during data collection were excluded from the study and represented by other participants during data collection.

The study population included **914** under five children, 305 randomly selected from each region to represent ethnicity of each region using list of children from health extension workers of the selected areas as a sampling frame.

## Sample size and sampling procedure

To determine the diagnostic accuracy of MUAC in detecting undernutrition by taking weight for height (WHZ) as gold standard among under five children of different ethnic backgrounds in Ethiopia, the required sample size was determined for sensitivity and specificity of MUAC based on the following assumptions and formula for sensitivity and specificity sample size calculation [22]. $n = (1.96)^2 x \frac{0.95(1-0.95)}{0.05^2 * 0.24} = 305$

n = 1.96$^2$x (0.95x0.05)/0.05$^2$x0.24 = 3.8416x0.0475/0.0025x0.24 = 0.182476/ 0.0006 = 305Where;

n = expected sample size for sensitivity test, d = margin of error of = 0.05, SN = anticipated sensitivity = 0.95, SP = anticipated specificity = 0.95, P = expected national proportion underweight 24% from EDHS report of Ethiopia [23]. A = size of critical region = 5% 1-α = confidence level = 95% Z$_1$-α/2 = standard normal deviate corresponding to the specified size of critical region α = 1.96, $n = \left(\frac{Z\alpha}{2}\right)^2 x \frac{SN(1-SN)}{d^2 * p}$

**Sampling technique.** For selection of 915 under five children from study population, 305 from each region simple random sampling method was employed. The list of children obtained from health extension workers of the selected areas were used as sampling frames.

## Measurements

A data abstraction tool was prepared for anthropometric measurement weight, height and mid upper arm circumference (MUAC) as well as socio-demographic information like age of the child estimated from EPI card and birth certificate while ethnicity of the child was collected by interviewing the parents. The WHO standard measurement protocol was used for conducting anthropometric measurements [15]. Height was measured using standard wooden sliding portable stadiometer by keeping the head of the child in the Frankfurt plane with knees straight and the heels, buttocks and the shoulders blades touching the vertical stand of the stadiometer (anthropometry). For measuring weight seca Germany scale was used based on standard procedure, before each measurement the scale was calibrated to zero position and heavy clothes and shoes were removed. For measuring MUAC a non-stretchable insertion tape is typically

used and it's graduated in millimeters. The steps used during measurement; the non-dominant arm selected. Then arm of the child was bent at the elbow to a 90-degree angle to get mid-point, upper arm was held parallel to the side of the body, the arm length between the shoulder tip (acromion) and the point of the elbow (olecranon process) is measured and the midway identified and marked between these anatomical landmarks. The upper arm circumference is taken at this point, while the participant's arm hanged loosely by the side. The tape measure is comfortable around the upper arm but not too tight and the measurement is recorded to the nearest millimeter.

Four data collectors [24] and one supervisor were recruited based on their experience in data collection. Data collectors received a one-day training on anthropometric measurement and deployed to collect data once the principal investigator was convinced about their competency. The principal investigator of the study and the supervisors critically followed the data collection process to minimize missing information and inconsistencies.

## Operational definitions

Cohens Kappa results were interpreted as follows: values $\leq 0$ as indicating no agreement and 0.01–0.20 as none to slight, 0.21–0.40 as fair, 0.41–0.60 as moderate, 0.61–0.80 as substantial, and 0.81–1.00 as almost perfect agreement [25].

Wasting: weight-for-height Z-score $< -2$. It often indicates recent and severe weight loss, although it can also persist for a long time [20].

Moderate under nutrition (SAM): It is diagnosed by weight for- height below -2SD of the WHO standards, by a MUAC $<12.5$ cm and it includes stunting or low length/height-for-age, wasting or low weight-for length/ height or bilateral pitting edema and underweight or low weight-for-age [21,23].

## Data processing and analysis

Data were coded, entered into Ep-data version 4.2 software and exported to SPSS for windows version 25 software for analysis. The presence of missing values, possible outliers, and multi-collinearity were checked through exploratory analysis. Before main analysis inter-rater reli-ability of WHZ and MUACZ was checked with Cohen's Kappa analysis for agreement between test variable MUAC and state variable WHZ$<$-2 and sensitivity and specificity as well as Kappa value were determined. A MUAC value $<12.5$ was used to define moderate acute malnutrition, while in similar direction of definition for moderate acute malnutrition was used by the gold standard WHZ$<$-2 [9].

Finally, cut off point for MUAC was developed using receiver operating characteristic (ROC curve) based on area covered under the curve using WHZ$<$-2 score as gold standard binary classifier.

**Ethical considerations.** Before starting the data collection process, the study was ethically approved by Jimma University Health Research Ethics Review Committee (IHRERC). An official letter was written from Jimma University to each regional, namely Gambella, Amhara and Somalia Health Office. Informed written assents was obtained from all parents of selected under five children and confidentiality of the study documents and the abstracted information was ensured according to the ethical principles enshrined in the Helsinki declaration.

## Results

In this study, 914 under five children were involved from three regions namely from Gambella, Amhara, Somalia regions representing different ethnic groups. In the three regions, children from four ethnic groups were involved in the study constituting Anuak (16.3%), Nuer (17.1%),

Amhara (33.4%) and Somalia (33.3%) approximately half of (50.8%) them were males and 49.2% were females (Table 1).

## Nutritional status of under-five children

In this study 9.7% under five children were malnourished for WHZ<-2 and 6% were malnourished for MUAC<12.5cm. Concerning to regional or ethnic differences in in the agreement of malnutrition by WHAZ and MUAC, in Amhara region, a prevalence were 16.4% and 3.6% were observed WHZ < - 2 Z and MUAC < 12.5 cm, respectively and in Gambella region 23.6% were malnourished for WHZ<-2 and only 5.9% of under-five children's malnourished for MUAC<12.5cm. Similarly, for Somali region, prevalence of 19.1% and 8.6% were documented by WHZ < - 2 Z and MUAC < 12.5 cm, respectively.

Regarding gender differences in the diagnostic agreement between WHZ and MUAC, a moderate wasting prevalence of 22.0% and 5.8% using WHZ < - 2 Z and MUAC < 12.5 cm, respectively among males. Likewise, a prevalence of 17.3% and 6.2% had moderate wasting using WHAZ<-2 and MUAC<12.5cm, among females under five children's respectively (Table 2).

## Estimating inter-rater reliability of WHZ and MUACZ with Cohen's Kappa

Inter rater reliability for diagnosis of under five children for Moderate acute malnutrition was using Cohen's Kappa for the overall sample from the three regions (Gambella, Amhara and Somalia). Sensitivity of MUAC<12.5cm in screening of undernutrition was 20.6% while its specificity was 97.5% showing a fair agreement with (Kappa = 0.0245, P = 0.001).

For Gambella region (among Nuer and Anuak ethnic groups) the sensitivity and specificity of MUAC<12.5cm were 16.7% and 97.4%, respectively showing a slight agreement of (Kappa = 0.19, p = 0.001). In similar way, for Somalia region (among Somali ethnic groups) sensitivity and specificity of MUAC were 29.3% and 96.3% with fair agreement (Kappa = 0.325, p = 0.001).

Equally, for Amhara region sensitivity and specificity of MUAC<12.5cm were 16.0% and 98.8%, respectively showing a fair agreement (Kappa = 0.216, P = 0.001) (Table 3).

**Table 1. Socio demographic characteristics of study participants from three regional states of Ethiopia N = 914.**

| Variable | | Frequency (%) |
|---|---|---|
| Sex | | |
| Male | | 464(50.8) |
| Female | | 450(49.2) |
| Age in months | | |
| 6–11 | | 66(7.2) |
| 12–23 | | 106(11.6) |
| 24–35 | | 119(13.) |
| 36–47 | | 262 (28.7) |
| 48–60 | | 361(39.5) |
| Region | | |
| Gambella | | 305(33.4) |
| Amhara | | 305(33.4) |
| Somalia | | 304(33.3) |
| Ethnic combination | | |
| Anuak | | 149(16.3) |
| Nuer | | 156(17.1) |
| Amhara | | 305(33.4) |
| Somalia | | 304(33.3) |

**Table 2. Nutritional status of under-five children's by using WHZ and MUACZ from different regions of Ethiopia (n = 914).**

| Variables | Measurement by WHZ <-2 | | Measurement by MUAC< 12.5cm | | Percent misclassified by MUAC<12.5 |
|---|---|---|---|---|---|
| | Normal | Malnourished | Normal | Malnourished | |
| | n (%) | n (%) | n (%) | n (%) | |
| **Region** | | | | | |
| Gambella | 233(76.4) | 72(23.6) | 287(94.1) | 18(5.9) | 17.7 |
| Amhara | 255(83.6) | 50(16.4) | 294(96.4) | 11(3.6) | 12.8 |
| Somalia | 246(80.9) | 58(30) | 278(91.4) | 26(8.6) | 10.5 |
| Total | 734(80.3) | 180(19.7) | 859(94.0) | 55(6.0) | 13.7 |
| **Age in month** | | | | | |
| 6–11 | 56(84.8) | 10(15.2) | 61(92.4) | 5(7.6) | 7.6 |
| 12–23 | 94(88.7) | 12(11.3) | 98(91.6) | 9(8.4) | 1.9 |
| 24–35 | 93(78.2) | 26(21.8) | 108(90.0) | 12(10.0) | 11.8 |
| 36–47 | 215(82.1) | 47(17.9) | 251(95.8) | 11(4.2) | 13.7 |
| 48–60 | 276(76.5) | 85(23.5) | 341(95.0) | 18(5.0) | 18.5 |
| **Sex** | | | | | |
| Male | 362(78.0) | 102(22.0) | 437(94.2) | 27(5.8) | 16.2 |
| Female | 372(82.7) | 78(17.3) | 422(93.8) | 28(6.2) | 11.1 |

## Optimal cut off point of MUAC for detection of moderate acute malnutrition

Cut-off point for MUAC that was equivalent with WHZ<-2 score as gold standard was determined using Receiver Operating Characteristic (ROC curve). Based on ROC analysis, the cut-off point of MUAC for the overall sample involving children from the three regions (Gambella, Amhara and Somali) 13.85cm with sensitivity and specificity of 98,9% and 80.7, respectively (P = 0.001).

There was a regional difference in the new MUAC cut-off, sensitivity and specificity in diagnosing moderate acute malnutrition. In Gambella Region, an optimal MUAC cut-off was 13.85 cm with sensitivity and specificity of 98.6 and 81.7, respectively. This value corresponds to the positive likelihood ratio of 5.4 and the area under roc curve of 0.92. For Amhara Region, the optimal MUAC cut-off for diagnosing MAM was 13.85 cm with sensitivity and specificity of 100% and 87.5%, respectively giving the corresponding positive likelihood ratio of 8 and an area under roc curve of 0.97. For Somali Region, the optimal MUAC cut-off for diagnosing MAM was 13.75 cm with a sensitivity and specificity of 98.3% and 72.9%, respectively giving the corresponding positive likelihood ratio of 3.6 and area under roc curve of 0.90. (Table 4, Figs 1, 2, 3 and 4).

**Table 3. Validity of MUAC in detecting moderate acute malnutrition among different ethnic groups of Ethiopia as compared to as compared to weight for Height Z score as gold standard.**

| Region | Test tool (MUAC<12.5 | Standard tool (Weight for height Z score or WHZ < -2) | | | | | | | | | | | |
|---|---|---|---|---|---|---|---|---|---|---|---|---|---|
| | | TP(a) | FP(b) | FN(c) | TN(d) | Total | Sensitivity (%) | Specificity (%) | PPV (%) | NPV (%) | Kappa | Agreement | P |
| Gambella | MUAC <12.5 cm | 12 | 6 | 60 | 227 | 305 | 16.7 | 97.4 | 66.7 | 79.1 | 0.19 | Slight | <0.001 |
| Somali | MUAC <12.5 cm | 17 | 9 | 41 | 237 | 304 | 29.3 | 96.3 | 65.4 | 85.3 | 0.325 | fair | <0.001 |
| Amhara | MUAC <12.5 cm | 8 | 3 | 42 | 252 | 305 | 16.0 | 98.8 | 72.7 | 85.7 | 0.216 | fair | <0.001 |
| Overall | MUAC <12.5 cm | 37 | 18 | 143 | 716 | 914 | 20.6 | 97.5 | 67.3 | 16.6 | 0.245 | fair | <0.001 |

Sensitivity = a/a + c, specificity = d/b + d, positive predictive value (PPV) = a/a + b, negative predictive value (NPV) = d/c + d.

Kappa agreement (0 = no/poor), (0.01–0.20 = slight), (0.21–0.40 = fair), (0.4–0.60 = moderate), (0.61–0.80 = substantial), and (0.81–1.00 = almost perfect).

**Table 4. Optimal MAUC Cut off for moderate acute malnutrition among under five children using weight for height Z<-2 as a gold standard.**

| Region | Optimal MUAC Cut-off | Sensitivity | Specificity | Youden index | Positive Likelihood Ratio | Area under curve | P | CI at 95% |
|---|---|---|---|---|---|---|---|---|
| Over all | 13.85 | 0.99 | 0.80 | 0.79 | 4.99 | 0.93 | 0.001 | 0.91–0.94 |
| Gambella | 13.85 | 0.99 | 0.81 | 0.80 | 5.20 | 0.92 | 0.001 | 0.89–0.95 |
| Amhara | 13.85 | 1.00 | 0.87 | 0.87 | 7.70 | 0.96 | 0.001 | 0.94–0.98 |
| Somalia | 13.75 | 0.98 | 0.73 | 0.71 | 3.60 | 0.90 | 0.001 | 0.88–0.94 |

## Discussion

In this study we identified significant variation in diagnosis of moderate acute malnutrition by MUAC<12.5cm and WHZ<-2-score among under five children in the overall sample as well as in each of the three regions. In the overall sample, the prevalence of moderate acute malnutrition was estimated to be 19.7% by WHZ<-2, whereas it was 6.0% by MUAC<12.5 cm indicating a misclassification of 13.7%.

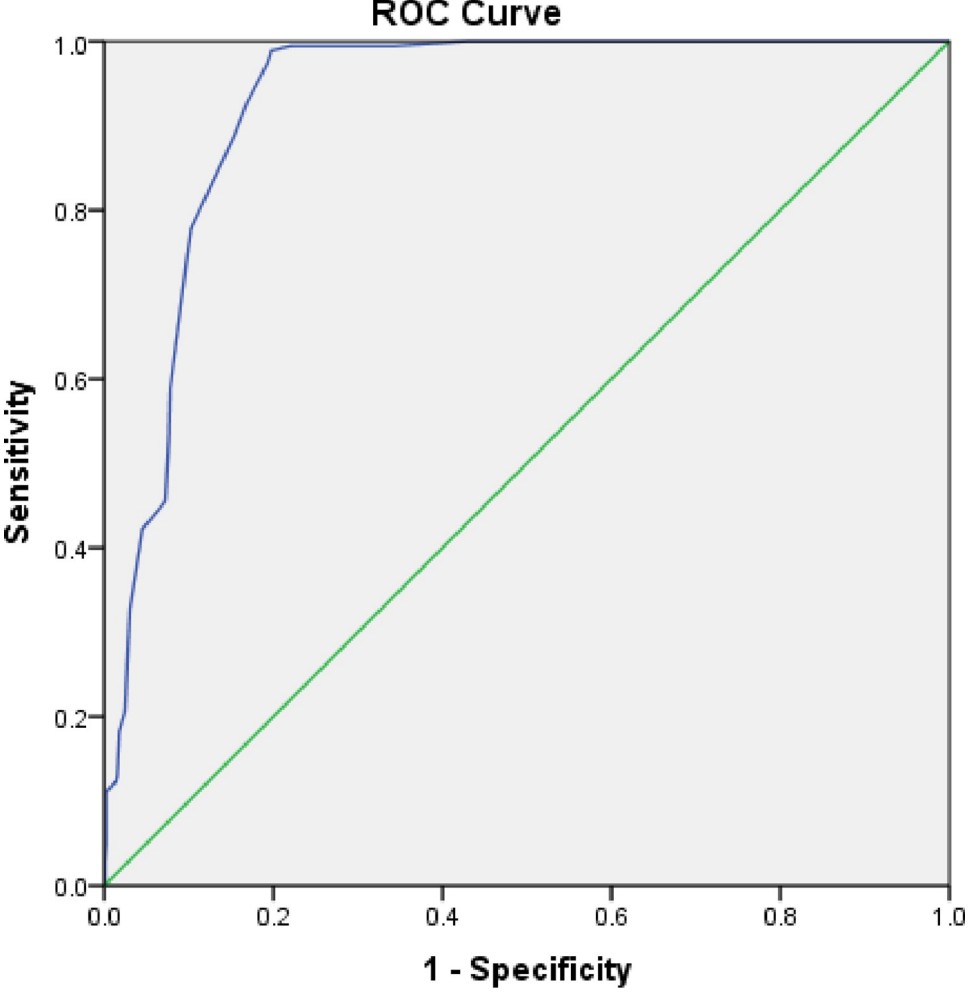

**Fig 1. ROC curve for sensitivity and specificity of MUAC as compared to WHZ<-2 among under five children of Ethiopia.**

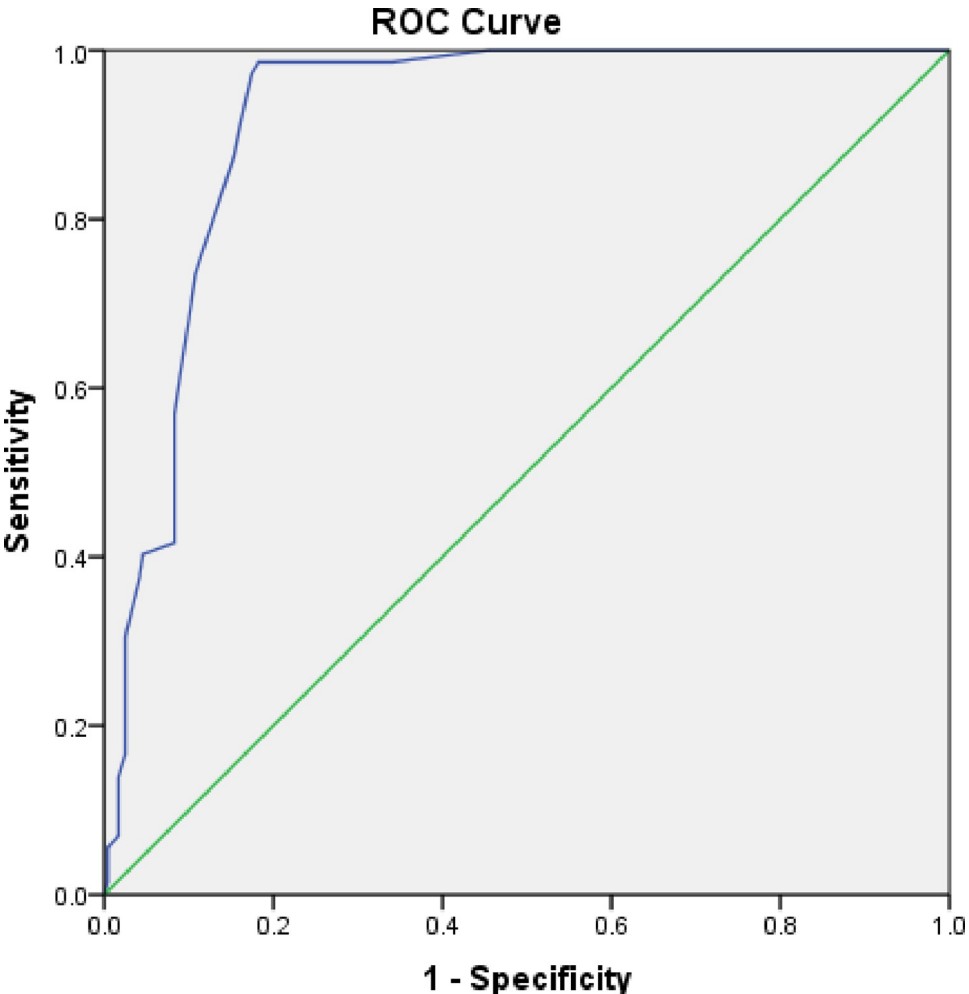

**Fig 2. ROC curve for sensitivity and specificity of MUAC as compared to WHZ<-2 among under five children of Gambella, Ethiopia.**

This difference also varies among different ethnic groups across region. In Gambella Region among Nuer and Anuak ethnicity the prevalence of moderate acute malnutrition was 23.6% by WHZ<-2 score and 5.9% by MUAC<12.5 cm showing difference of 17.7%. For Amhara region, the prevalence of MAM was 16.4% by WHZ<-2 while it was 3.6% by MUAC<-12.5 showing a difference of 12,8%. Likewise, for Somalia Region the prevalence of MAM was 19.1% by WHZ<-2 and 8.6% by MUAC<12.5cm resulting in a difference of 6.6%.

The findings show that the current MUAC cut-off (<12.5cm) used in real service setup for screening children significantly underestimates the prevalence of MAM leading to a significant misclassification as compared to WHZ<-2 for the whole country although there were some regional variations. The highest misclassification (17.7%) was observed in Gambella Region while the lowest misclassification (6.6%) was observed in Somali Region.

The sensitivity and specificity of the current MUAC cut-off <12.5cm as compared to WHZ <-2, for the whole sample were 20.6% and 97.5%, respectively with fair agreement (Cohen's Kappa = 0.245) with WHZ<-2 score, which is consistent with a study in India that showed a

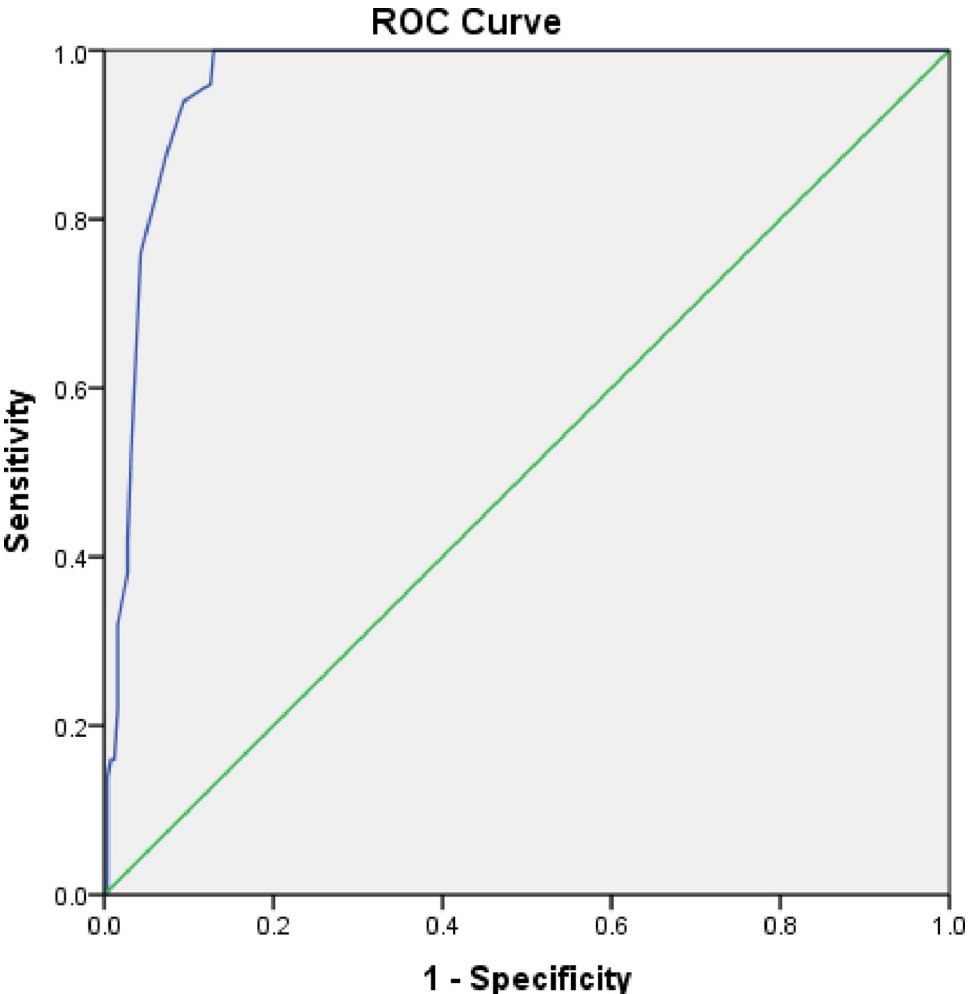

**Fig 3. ROC curve for sensitivity and specificity of MUAC as compared to WHZ<-2 among under five children of Amhara, Ethiopia.**

low sensitivity and specificity of this MUAC Cut-off [20]. This finding implies an urgent need for revising the cutoff value of MUAC to higher value to improve its sensitivity in detecting children with MAM. The Cohen's Kappa result also suggests the need for developing a new cut off for health intervention programs better to have maximum Cohen's Kappa [25].

There was a regional difference the diagnostic validity of the currently MUAC cut-off <12.5cm as compared to WHZ<-2. For Gambella region among Nuer and Anuak ethnic groups it showed a lower sensitivity (16.7%) and higher specificity (97.4%) with slight agreement (Kappa = 0.19), suggesting that the existing cut off point of MUA<12.5 cm for detection of under nutrition may not capture all under nourished children. Therefore, we need to have new cut off point to maximize success of the treatment and to minimize the relapse of malnourished cases.

Similarly, for Amhara region, the existing MUAC cut off point for diagnosing MAM (<12.5cm) had lower in sensitivity (16.0%) and high specificity (98.8%) giving a fair agreement (Kappa-value = 0.216) with WHZ<-2 score. This also needs revision of the cut-off to higher

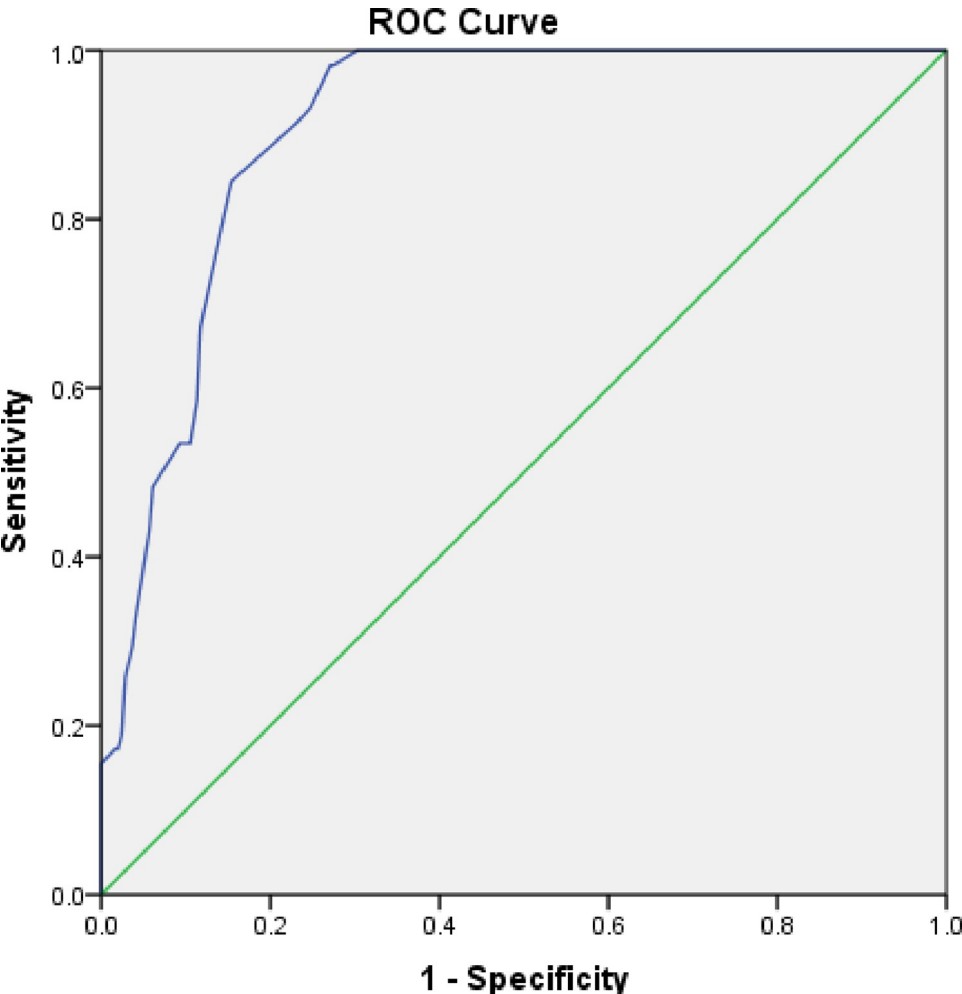

**Fig 4. ROC curve for sensitivity and specificity of MUAC as compared to WHZ<-2 among under five children of Somalia, Ethiopia.**

values to maximize detection of MAM cases and improve inter reliability agreement with gold standard.

In similar way, of the diagnostic performance of the currently used MUAC cut-off (<12.5cm) showed low sensitivity (29.3%) and high specificity (96.3%) for detection of MAM among the children from the Somalia region or Somali ethnic group showing a faire agreement (Kappa- = 0.325) compared to gold standard WHZ<-2 score. This result also shows the currently used MUAC cut-off fails to detect significant proportion of MAM cases Somali Region implying the need for revision of the MUAC cut-off for a better diagnostic performance.

Based on the findings of low performance of the existing MUAC in diagnosing MAM cases, new cut off points of MUAC were developed using ROC curve for the overall sample as well as for the three regions. Finding showed that the revised MUAC cut-off for the overall sample (three regions namely for Gambella, Amhara and Somalia) 13.85cm with better sensitivity (98.9%) and specificity (80.7%) with the positive likelihood ratio of 5.1 and area under ROC curve of 0.93 showing an excellent performance.

For Gambella and Amhara regions the optimal cut-off was similar to the total sample (MUAC<13.85) giving good sensitivity (96.6%) and specificity (81.7%). This result shows that this MUAC cutoff point enhances the diagnostic performance for identifying MAM cases and prevent relapse. However, the optimal cutoff point of MUAC for diagnosis of MAM was MUAC<13.75cm for Somali Region giving highest sensitivity (98.3%) and specificity (72.9). The suggestions of WHO growth standards confirm prior explanations that the effect of ethnic variations on the growth of infants and young children in populations is small compared with the effects of the environment but there are genetic differences among individuals [26]. The difference in the optimal cut-off value for Somali region may be explained due to differences in environmental conditions [27] and may be by differences in body frames. The findings of this study have wider practical implications. The fact that the currently used MUAC cut-off demonstrated low performance in diagnosing MAM cases and that the higher optimal cut-off (MUAC <13.85) and finding in this study were parallel with study conducted in Cambodia and Nepal [28,29] this study suggests that the need for revision of the cut to 13.85 to enhance the diagnostic capacity of the MUAC as screening tool. As wasting (Moderate acute malnutrition) is associated with stunting if it happens for prolonged duration, failure to diagnose and treat MAM cases timely due to the use of the current MUAC cut-off(<12.5cm) would enhance relapse cases and becomes a stumbling block of the efforts to reduce stunting to zero level.

This study involved different regions with different ethnic backgrounds and geographical locations. It gives a new insight for cutoff point of MUAC among different regions and ethnic variation. However, it did not take sample from other ethnic groups, although much of as variability is not expected from the study regions represented due to similarities of the majority of the ethnic groups in body frames to either of the ethnic groups represented in sample from the three regions.

## Conclusion

Findings from this study suggests that current MUAC cut-off of <12.5cm cutoff point is less sensitive for diagnosis of MAM and significantly underestimates the caseload, relapse and consequent chronic malnutrition. And the optimal cut-off that gave high sensitivity and specificity for diagnosing MAM cases <13.85cm.

**Policy implications.** Based on the findings, it is recommended that federal ministry of health expected to revise the MUAC cut-off point to <13.85cm to enhance its diagnostic performance.

**Limitation.** This study is mainly limited to three regions because of shortage of resources, as we know that Ethiopia is home for diversity with 10 regions and this may limits to conclude this study at national level.

**Recommendation.** Ethiopia is home of diverse ethnic groups with different body compositions, geographical and climate conditions there was need to validate and develop new cut off points for each region among different ethnic groups. Better to conduct further investigation among different ethnic groups and different regions in Ethiopia.

## Supporting information

**S1 File.**
(RAR)

## Acknowledgments

I want acknowledge my data collectors Habtamu Bekala and his team Gambella health office and Keliab Sisay and his team Somalia region health office for their commitment for valuable

data extraction, Tsione Hailu for data feeding to software and at last but not the least I want to thank my mother Lero Biramo for her advice on my nonacademic work that can have effect on my academic performance.

## Author Contributions

**Conceptualization:** Abera Lambebo, Dessalegn Tamiru, Tefera Belachew.

**Data curation:** Abera Lambebo, Yordanos Mezemir, Tefera Belachew.

**Formal analysis:** Abera Lambebo, Dessalegn Tamiru, Tefera Belachew.

**Funding acquisition:** Abera Lambebo, Dessalegn Tamiru, Tefera Belachew.

**Investigation:** Abera Lambebo.

**Methodology:** Abera Lambebo, Dessalegn Tamiru, Tefera Belachew.

**Project administration:** Abera Lambebo, Tefera Belachew.

**Resources:** Abera Lambebo, Yordanos Mezemir, Tefera Belachew.

**Software:** Abera Lambebo, Dessalegn Tamiru, Tefera Belachew.

**Supervision:** Yordanos Mezemir, Tefera Belachew.

**Validation:** Abera Lambebo, Tefera Belachew.

**Visualization:** Tefera Belachew.

**Writing – original draft:** Abera Lambebo, Tefera Belachew.

**Writing – review & editing:** Abera Lambebo, Dessalegn Tamiru, Tefera Belachew.

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
