## [Decision Letter · Decision Letter 0]

18 Feb 2022

PONE-D-21-31067Validating the Diagnostic Performance of MUAC in screening moderate Acute Malnutrition and developing an optimal cut-off for Under Five Children of Different RegionsPLOS ONE

Dear Dr. Lambebo,

Thank you for submitting your manuscript to PLOS ONE. After careful consideration, we feel that it has merit but does not fully meet PLOS ONE’s publication criteria as it currently stands. Therefore, we invite you to submit a revised version of the manuscript that addresses the points raised during the review process.

We look forward to receiving your revised manuscript.

Kind regards,

Ammal Mokhtar Metwally, Ph.D (MD)

Academic Editor

PLOS ONE

Journal Requirements:

No;The funders had no role in study design, data collection and analysis, decision to publish, or preparation of the manuscript.

Additional Editor Comments:

Please note that your manuscript was reviewed by 2 experts in the field. There is consensus agreement that the idea of the article is interesting. Meanwhile, some of the reviewers identified important problems in your submission and provided copious comments. It is also required to consider adding more references from countries with similar context (low- and middle-income countries) in the discussion section.

The manuscript could be greatly strengthened by considering editing according to the specific mentioned comments.

Please note that further language improvement is indicated. Consider revising the spelling, grammar, diction, and syntax throughout the manuscript for increased clarity.

Reviewers' comments:

Reviewer's Responses to Questions

**Comments to the Author**

1. Is the manuscript technically sound, and do the data support the conclusions?

Reviewer #1: Yes

Reviewer #2: Partly

2. Has the statistical analysis been performed appropriately and rigorously? 

Reviewer #1: Yes

Reviewer #2: Yes

3. Have the authors made all data underlying the findings in their manuscript fully available?

Reviewer #1: Yes

Reviewer #2: Yes

4. Is the manuscript presented in an intelligible fashion and written in standard English?

Reviewer #1: Yes

Reviewer #2: Yes

5. Review Comments to the Author

Reviewer #1: Dear the authors of this mnuscript you have done a great work, but there are some comments to make things better:

Regarding the language: some verbs in your manuscript were used in the ppresent tense, you should put them in the past as the others. The grammar needs to be checked.

- Title: you should mention where are these different regions????

-Introduction:1- in the line before the last one in the first paragraph you mentioned SAM please mention the full word. 2- you should define what is malnutrition and classification and degrees of it.

3- you should also mention the prevalence of malnutrition globally and the magnitude of the problem in your country, and why did you choose these four regions for your research; is the prevalence of malnutrition more in these regions than the other parts of the country.

4- Please clarify; what did you mean by the sentence; " In addition to that the currently existing tools have low treatment outcome, increasing number of relapse and overburdening health system by relapse cases".

-Methods: 1-you mentioned that you selected three different ethnicities, but you mentioned that one of the three regions which is Gambella has two different ethnic groups so by this you had 4 different ethnic groups as you mentioned here; "For this study Gambella Town used to represent two ethnic groups those were Nuer and Anuak to represent Nailo Saharans’ linguistic groups. So I think in your results and dicussion you should analyse the results of those regions separately and don't join them together in one area which is Gambella.

2-in the paragraph started by Somali regional state: in the last line you mention the word (groups) twice so remove one of them.

3- you mentioned that, All apparently healthy under five children who were members of

the community for more than six months were (include) in the study, please explain why did you define 6 months as an inclusion criteria and please not that the verb in the passive so replace it with (included).

4- you mentioned that, "For measuring weight hanging scale and seca Germany was used based on

standard procedure", I think the sentence is not correct as this is the first time I hear about a hanging scale used for humans so please define which scale was used.

5- you mentioned also in the above paragraph ,(heavy closes and shoes were removed). The word (closes) is incorrect in spelling, so remove it and put (clothes). and in the remaining part of this paragraph you used the verb (is) repeatedly, so please rvise the paragraph and put the verb (was).

7- in the operational definiotions; you defined wasting then you defined undernutrition as I think and as I understood from your the definitions they are the same so what did you mean by defining them separately.

- Results: 1- as regards the Nutritional status of under-five Children you need to revise the first paragraph. I think there are some words missing as you said, "Based on the definition of moderate acute malnutrition as WHZ<-2 and MUAC<12.5 cm, 9.7% and 6.0% were malnourished, respectively" I think you should mention that this was in the whole sample, Moreover the number (9.7%) regarding the prevalence detected by WHZ scores is not correct as you mentioned before and later in the tables of the results and in the discussion that it was (19.7%).

2- "Regarding regional or ethnic differences in in the agreement of malnutrition by WHAZ and MUAC, a prevalence of 23.6% and 5.9% were observed by WHZ < - 2 Z and MUAC < 12.5 cm, respectively". I think you forgot to say that this was in Gambella.

3- Regarding Gender difference " Likewise, a prevalence of 17.3% and 6.2% had moderate wasting using WHAZ<-2 and MUAC<12.5cm, respectively" I think you forgot to mention that this prevalence was in females.

-Discussion:

1- in the sentence "In Gambella Region among Nuer and Anuak ethnic" replace the word (ethnic) with (ethnicity).

2- in the sentence "Likewise, for Somalia Region the prevalence of MAM was 19.1% by WHZ<-2 and 8.6% by MUAC<12.5cm resulting in a misclassification of 6.6%." the misclassification is not = 6.6% as you mentioned but it's= 10.5%.

3- the same in the sentence"The highest misclassification (17.7%) was observed in Gambella Region

while the lowest misclassification (6.6%) was observed in Somali Region." the correct percent of misclassification is 10.5%.

4- in the sentence "Somali ethnic group showing a faire agreement (Kappa=0.325) compare to gold standard WHZ<-2 score" use compared instead of compare.

5- the sentence "of the findings showed that the revised MUAC cut-off for the overall sample" you should not begin a sentence with a preposition. Moreover, the whole sentence needs reformulation.

6- in the sentence " The difference in the optimal cut-of value for Somali region" correct the word (cut-of) repalc it by (cut-off).

7- in the sentence "higher optimal cut-off (MUAC <18.85) developed in this study" you should correct the value of cut-off point vas you mentioned before it was (MUAC< 13.85) and not 18.85.

-Conclusion: the sentence at the beginning of the paragraph "Findings current MUAC cut-off of <12.5cm cutoff point is less sensitive for diagnosis of MAM and significantly underestimates the caseload, relapse and consequent chronic malnutrition" needs reformulation.

-Recommendation:

1-in the beginning of the paragraph "The based on the findings" remove the word (The).

2-in the sentence "diverse ethnic" add the word (groups).

3- the sentence "different body composition, geographical and climate condition" use the words in plural; (compositions, conditions).

-In general the discussion is defficient in comparisons with other studies performed either in the same regions or allover the country or even worldwide.

Reviewer #2: The following suggestions are recommended:

It will be logical to spell out the MUAC abbreviation early in the abstract as some readers might not be aware of this.

Introduction: The main aim of the study should be highlighted in the end as a separate brief paragraph.

Discussion: The limitations of the study should be clearly highlighted and the policy implications and future recommendations can also be further highlighted.

Typo and grammatical errors were also detected in the manuscript and a professional proof reading is advised.

6. PLOS authors have the option to publish the peer review history of their article (what does this mean?). If published, this will include your full peer review and any attached files.

Reviewer #1: No

Reviewer #2: No

---

## [Author Response · Author response to Decision Letter 0]

3 Mar 2022

all comments were accepted and corrected

thanks for all reviewers and editorials for your constructive comments.

---

## [Decision Letter · Decision Letter 1]

18 Apr 2022

PONE-D-21-31067R1Validating the Diagnostic Performance of MUAC in screening moderate Acute Malnutrition and developing an optimal cut-off for Under Five Children of Different RegionsPLOS ONE

Dear Dr. Lambebo,

Thank you for submitting your manuscript to PLOS ONE. After careful consideration, we feel that it has merit but does not fully meet PLOS ONE’s publication criteria as it currently stands. Therefore, we invite you to submit a revised version of the manuscript that addresses the points raised during the review process.

We look forward to receiving your revised manuscript.

Kind regards,

Ammal Mokhtar Metwally, Ph.D (MD)

Academic Editor

PLOS ONE

Reviewers' comments:

Reviewer's Responses to Questions

**Comments to the Author**

1. If the authors have adequately addressed your comments raised in a previous round of review and you feel that this manuscript is now acceptable for publication, you may indicate that here to bypass the “Comments to the Author” section, enter your conflict of interest statement in the “Confidential to Editor” section, and submit your "Accept" recommendation.

Reviewer #1: (No Response)

2. Is the manuscript technically sound, and do the data support the conclusions?

Reviewer #1: Yes

3. Has the statistical analysis been performed appropriately and rigorously? 

Reviewer #1: Yes

4. Have the authors made all data underlying the findings in their manuscript fully available?

Reviewer #1: Yes

5. Is the manuscript presented in an intelligible fashion and written in standard English?

Reviewer #1: No

6. Review Comments to the Author

Reviewer #1: Dear the authors of this manuscript, you have done a great work but you didn' address most of my comments. So I think you need to revise the manuscript again accurately and read all my previous comments accurately and correct the mistakes to make your manuscript appear in the best and complete correct form.

Regarding the language there are many grammatical and typographical errors so I advise you to search copyediting service and get professional proof reading. In addition the results needs to be revised regarding some numbers and percents mistakes. Moreover, the discussion is still poor and deficient in comparisons with other studies. you must retype the conclusion and recommendations also.

you didn't mention the limitations of the study.

Finally, as I mentioned before you have already addressed some of my comments of the last revision adequately but most of them were addressed partially or inadequately or were not addressed at all.

7. PLOS authors have the option to publish the peer review history of their article (what does this mean?). If published, this will include your full peer review and any attached files.

Reviewer #1: No

---

## [Author Response · Author response to Decision Letter 1]

18 Apr 2022

all comments by reviewer 2 were accepted and corrected.

---

## [Decision Letter · Decision Letter 2]

12 Jun 2022

PONE-D-21-31067R2Validating the Diagnostic Performance of MUAC in screening moderate Acute Malnutrition and developing an optimal cut-off for Under Five Children of Different RegionsPLOS ONE

Dear Dr. Lambebo,

Thank you for submitting your manuscript to PLOS ONE. After careful consideration, we feel that it has merit but does not fully meet PLOS ONE’s publication criteria as it currently stands. Therefore, we invite you to submit a revised version of the manuscript that addresses the points raised during the review process.

We look forward to receiving your revised manuscript.

Kind regards,

Ammal Mokhtar Metwally, Ph.D (MD)

Academic Editor

PLOS ONE

Journal Requirements:

Additional Editor Comments (if provided):

A great effort was made by the authors to utilize the feedback that was provided for them to correct. I find it interesting and improved with respect to the original submission. Please consider responding to the mentioned comments by the reviewers

Reviewers' comments:

Reviewer's Responses to Questions

**Comments to the Author**

1. If the authors have adequately addressed your comments raised in a previous round of review and you feel that this manuscript is now acceptable for publication, you may indicate that here to bypass the “Comments to the Author” section, enter your conflict of interest statement in the “Confidential to Editor” section, and submit your "Accept" recommendation.

Reviewer #1: (No Response)

Reviewer #3: All comments have been addressed

2. Is the manuscript technically sound, and do the data support the conclusions?

Reviewer #1: Yes

Reviewer #3: Yes

3. Has the statistical analysis been performed appropriately and rigorously? 

Reviewer #1: Yes

Reviewer #3: Yes

4. Have the authors made all data underlying the findings in their manuscript fully available?

Reviewer #1: Yes

Reviewer #3: Yes

5. Is the manuscript presented in an intelligible fashion and written in standard English?

Reviewer #1: No

Reviewer #3: Yes

6. Review Comments to the Author

Reviewer #1: Dear the authors of this manuscript you have done a great work but till there are some corrections needed to make things better:

- There are still some grammatical and typographical mistakes, so you need to review the manuscript by an English native person.

- Regarding the Introduction:

1- You have to notice that some readers may not be specialised so you should give an idea about malnutrition; its definition, classification, diagnosis and complications; even if if your main topic is about tools.

2- You didn't mention the prevalence of malnutrition globally and the magnitude of the problem in your country.

3- in line 99, (this study aimed to determine the diagnostic performance of MUAC for determination

of moderate wasting among children of different ethnic background and develop optimal cut-off.); Please use (acute malnutrition among under five children) instead of (moderate wasting among children) and add (s) to (background)

4-in line 100, please add the words (new) and (point) at the sentence to become (new optimal cut-off point)

- Regarding the Methods:

1- in line 128, you mentioned that the number of patients was (914) while you said that from each of the three regions you have done 305 cases so the total number must be (915).

2- in line 155, please add the word (scale) to become (Seca Germany Scale was used).

- Regarding the Results:

1- in line 202, you should add the word (Approximately) to (half of them) as 50.8% is not exactly the half of cases.

2- as regards the Nutritional status of under-five Children you need to revise the first paragraph. I think there are some words missing as you said, "Regarding regional or ethnic differences in in the agreement of malnutrition by WHAZ and MUAC, a prevalence of 23.6% and 5.9% were observed by WHZ < - 2 Z and MUAC < 12.5 cm, respectively". I think you forgot to say that this was in(Gambella).

- Regarding the Discussion:

1- in line 254, the difference is not 6.6 as 19.1 - 8.6 = 10.5

2- in line 259, the same previously mentioned mistake; as you said (the lowest misclassification 6.6 was observed in Somali region) while the correct is 10.5.

3- in line 301, remove the word (and) & replace the word (finding) with (found) and use (two studies) instead of (study).

4- in line 302, replace (this study suggests) with (which suggested).

5- in line 302, you mentioned ( revision of the cut to 13.85), I think you forgot to write (-off point) to be (cut-off point).

6- in line 305, (would enhance relapse cases) add (of) before (cases).

7- in line 306, replace ( becomes) with (would become).

8- in line 307, (this study involved different regions) add (in Ethiopia).

9- in line 309, add (s) to (variation) and to (sample) to be in plural.

10- in line 309 till line 312, the sentrence started with (however) till (regions) is not written accurately and its meaning is not clear, so it needs reformulation.

- Regarding the conclusion:

1- in line 313, remove the (s) from te verb (suggests) as the word (Findings) is in plural, you repeated the words (cut-off) so add (point) to (MUAC cut-off <12.5 cm) and remove (cut-off point) from (< 12.5 cut-off point).

2- in line 315, remove the word (And) as you can't start a sentence with And. Add the word (point) to (cut-off).

3- in line 316, add (was) to be (MAM cases was < 13.85 cm) .

- Regarding Policy implications:

1- in line 317, add (of our study) to (based on the findings).

- Regarding limitations of the study:

1- in line 320, remove (s) from (this may limits).

2- replace (conclude) with (conduct) and put the sentence like that (and to conduct this study at the national level, more fund is needed).

- Regarding the Recommendations:

1- in line 323, start the Sentence with (As) so the sentence will begin by (As Ethiopia is) and put a comma before (there was need), add (Anthropometric) before (cut-off points) and add (for early detection of Malnutrition) after (develop new cut off points).

2- in line 324, start with (It's better to conduct) and add (s) to (investigation) to be in plural.

Reviewer #3: I would like to thank authors for there very useful research. After corrections manuscript has improved a lot. Globally well understood.

7. PLOS authors have the option to publish the peer review history of their article (what does this mean?). If published, this will include your full peer review and any attached files.

Reviewer #1: No

Reviewer #3: No

---

## [Author Response · Author response to Decision Letter 2]

20 Jun 2022

comments given by reviewer 1 is corrected and accepted at response later

---

## [Decision Letter · Decision Letter 3]

19 Jul 2022

PONE-D-21-31067R3Validating the Diagnostic Performance of MUAC in screening moderate Acute Malnutrition and developing an optimal cut-off for Under Five Children of Different Regions in EthiopiaPLOS ONE

Dear Dr. Lambebo,

Thank you for submitting your manuscript to PLOS ONE. After careful consideration, we feel that it has merit but does not fully meet PLOS ONE’s publication criteria as it currently stands. Therefore, we invite you to submit a revised version of the manuscript that addresses the points raised during the review process.

We look forward to receiving your revised manuscript.

Kind regards,

Ammal Mokhtar Metwally, Ph.D (MD)

Academic Editor

PLOS ONE

Journal Requirements:

Additional Editor Comments (if provided):

Great effort was made by the authors to utilize the feedback that was provided for them to correct their manuscript. I find it interesting and improved with respect to the original submission. Please consider responding to the reviewers’ remarks. The manuscript could be greatly strengthened by considering editing according to the specific mentioned comments.

Reviewers' comments:

Reviewer's Responses to Questions

**Comments to the Author**

1. If the authors have adequately addressed your comments raised in a previous round of review and you feel that this manuscript is now acceptable for publication, you may indicate that here to bypass the “Comments to the Author” section, enter your conflict of interest statement in the “Confidential to Editor” section, and submit your "Accept" recommendation.

Reviewer #3: All comments have been addressed

Reviewer #4: (No Response)

2. Is the manuscript technically sound, and do the data support the conclusions?

Reviewer #3: Yes

Reviewer #4: Yes

3. Has the statistical analysis been performed appropriately and rigorously? 

Reviewer #3: Yes

Reviewer #4: Yes

4. Have the authors made all data underlying the findings in their manuscript fully available?

Reviewer #3: Yes

Reviewer #4: Yes

5. Is the manuscript presented in an intelligible fashion and written in standard English?

Reviewer #3: Yes

Reviewer #4: No

6. Review Comments to the Author

Reviewer #3: I must thank all authors for their effort. Authors have address all the review comments and manuscript has been improved significantly.

Reviewer #4: General comment

The paper introduces a new anthropometric cut-off of underweight nutritional status in children under the age of 5 in Ethiopia. The topic of the manuscript is of high importance, since underweight nutritional status in children represents an epidemic problem in the studied region in Africa. I recommend to accept this paper for publication in PlosOne after a minor revision. The main reasons of this revision request are:

• I miss information on why the same cut-off value of a body dimension can be reliable between 0-5 years of age for screening undernutrition status, please clarify it in the Introduction section that such a method, only one cut-off of MUAC can be used in this age interval. It should also be clarified why MUAC and not weight to height cut-off or z-scores are suggested for underweight screening in children under the age of 5 in the studied region of Africa.

• It is not clear how the use of MUAC cut-off value can distinguish between acute and chronic MAM. My suggestion is not to use the ‘acute’ attributive of MAM – either in the abstract or in the whole manuscript. Only one screening examination cannot define whether it is acute or chronic MAM in children, especially when we use only one anthropometric dimension.

• The manuscript needs extensive revision for language and grammar.

I list my specific/minor comments and suggestions to the manuscript in the order of the chapters of the manuscript to help the revision:

Abstract

A1: “Test variables is mid upper arm circumference (MUAC< 12.5cm).” – please correct this sentence, in this form it is not clear, too short, needs addition to help understanding.

A2: “In Amhara region there was fair agreement between the two measures in diagnosing MAM …” – please define MAM abbreviation on its first use.

A4: “WHZ<-2 for diagnosing MAM was …” – please first define WHZ abbreviation (also in the main manuscript).

Introduction

I1: Line 73 – “Nutritional assessment can be done using anthropometric …” – change nutritional assessment to nutritional status assessment.

Discussion

D1: The Discussion section is too long, please shorten it.

7. PLOS authors have the option to publish the peer review history of their article (what does this mean?). If published, this will include your full peer review and any attached files.

Reviewer #3: No

Reviewer #4: **Yes: **Annamaria Zsakai

---

## [Author Response · Author response to Decision Letter 3]

25 Jul 2022

Given comments were addressed accordingly

---

## [Decision Letter · Decision Letter 4]

8 Aug 2022

PONE-D-21-31067R4Validating the Diagnostic Performance of MUAC in screening moderate Acute Malnutrition and developing an optimal cut-off for Under Five Children of Different Regions in EthiopiaPLOS ONE

Dear Dr. Lambebo,

Thank you for submitting your manuscript to PLOS ONE. After careful consideration, we feel that it has merit but does not fully meet PLOS ONE’s publication criteria as it currently stands. Therefore, we invite you to submit a revised version of the manuscript that addresses the points raised during the review process.

We look forward to receiving your revised manuscript.

Kind regards,

Ammal Mokhtar Metwally, Ph.D (MD)

Academic Editor

PLOS ONE

Journal Requirements:

Additional Editor Comments (if provided):

Please note that the changes suggested by the reviewers are required before considering for publication. Please consider that wherever, there is an issue that need to be addressed –it should be addressed to meet PLOS one guidelines

Sorry Again for any inconvenience

Reviewers' comments:

Reviewer's Responses to Questions

**Comments to the Author**

1. If the authors have adequately addressed your comments raised in a previous round of review and you feel that this manuscript is now acceptable for publication, you may indicate that here to bypass the “Comments to the Author” section, enter your conflict of interest statement in the “Confidential to Editor” section, and submit your "Accept" recommendation.

Reviewer #3: All comments have been addressed

Reviewer #4: (No Response)

2. Is the manuscript technically sound, and do the data support the conclusions?

Reviewer #3: Yes

Reviewer #4: Yes

3. Has the statistical analysis been performed appropriately and rigorously? 

Reviewer #3: Yes

Reviewer #4: Yes

4. Have the authors made all data underlying the findings in their manuscript fully available?

Reviewer #3: Yes

Reviewer #4: No

5. Is the manuscript presented in an intelligible fashion and written in standard English?

Reviewer #3: Yes

Reviewer #4: No

6. Review Comments to the Author

Reviewer #3: (No Response)

Reviewer #4: I found some changes by following my comments in the Abstract, thank you.

Howevere, I did not get reply to my 2 main questions either in the corrected manuscript or in the reply to the reviewer section:

• I miss information on why the same cut-off value of a body dimension can be reliable between 0-5 years of age for screening undernutrition status, please clarify it in the Introduction section that such a method, only one cut-off of MUAC can be used in this age interval. It should also be clarified why MUAC and not weight to height cut-off or z-scores are suggested for underweight screening in children under the age of 5 in the studied region of Africa.

• It is not clear how the use of MUAC cut-off value can distinguish between acute and chronic MAM. My suggestion is not to use the ‘acute’ attributive of MAM – either in the abstract or in the whole manuscript. Only one screening examination cannot define whether it is acute or chronic MAM in children, especially when we use only one anthropometric dimension.

As soon as I get reply to these 2 questions, I can review the revised (4) version.

7. PLOS authors have the option to publish the peer review history of their article (what does this mean?). If published, this will include your full peer review and any attached files.

Reviewer #3: No

Reviewer #4: **Yes: **Annamaria Zsakai

---

## [Decision Letter · Decision Letter 5]

12 Aug 2022

Validating the Diagnostic Performance of MUAC in screening moderate Acute Malnutrition and developing an optimal cut-off for Under Five Children of Different Regions in Ethiopia

PONE-D-21-31067R5

Dear Dr. Lambebo,

We’re pleased to inform you that your manuscript has been judged scientifically suitable for publication and will be formally accepted for publication once it meets all outstanding technical requirements.

Kind regards,

Ammal Mokhtar Metwally, Ph.D (MD)

Academic Editor

PLOS ONE

Additional Editor Comments (optional):

Reviewers' comments:

Reviewer's Responses to Questions

**Comments to the Author**

1. If the authors have adequately addressed your comments raised in a previous round of review and you feel that this manuscript is now acceptable for publication, you may indicate that here to bypass the “Comments to the Author” section, enter your conflict of interest statement in the “Confidential to Editor” section, and submit your "Accept" recommendation.

Reviewer #4: All comments have been addressed

2. Is the manuscript technically sound, and do the data support the conclusions?

Reviewer #4: Yes

3. Has the statistical analysis been performed appropriately and rigorously? 

Reviewer #4: Yes

4. Have the authors made all data underlying the findings in their manuscript fully available?

Reviewer #4: Yes

5. Is the manuscript presented in an intelligible fashion and written in standard English?

Reviewer #4: Yes

6. Review Comments to the Author

Reviewer #4: I checked the revised manuscript, although the Authors did not help my work to find the revisions in the text, as I see all my comments were accepted the revisions were done by following my comments and recommendations, too. Thank you!

7. PLOS authors have the option to publish the peer review history of their article (what does this mean?). If published, this will include your full peer review and any attached files.

Reviewer #4: **Yes: **Annamaria Zsakai

---

## [Editor Report · Acceptance letter]

20 Sep 2022

PONE-D-21-31067R5 

Validating the Diagnostic Performance of MUAC in screening moderate Acute Malnutrition and developing an optimal cut-off for Under Five Children of Different Regions in Ethiopia 

Dear Dr. Lambebo:

I'm pleased to inform you that your manuscript has been deemed suitable for publication in PLOS ONE. Congratulations! Your manuscript is now with our production department. 

Kind regards, 

on behalf of

Professor Ammal Mokhtar Metwally 

Academic Editor

PLOS ONE